# Executive Function and Attention Performance in Children with ADHD: Effects of Medication and Comparison with Typically Developing Children

**DOI:** 10.3390/ijerph16203822

**Published:** 2019-10-10

**Authors:** Martina Miklós, Judit Futó, Dániel Komáromy, Judit Balázs

**Affiliations:** 1Doctoral School of Psychology, ELTE Eötvös Loránd University, Izabella St. 46, 1064 Budapest, Hungary; 2Department of Developmental and Clinical Child Psychology, Institute of Psychology, ELTE Eötvös Loránd University, Izabella St. 46, 1064 Budapest, Hungary; futo.judit@ppk.elte.hu (J.F.); d.komaromy@student.vu.nl (D.K.); balazs.judit@ppk.elte.hu (J.B.); 3Department of Behavioral and Movement Sciences, Vrije Universiteit, De Boelelaan 1105, 1081 HV Amsterdam, The Netherlands; 4Department of Psychology, Bjørknes University College, Lovisenberggata 13, 0456 Oslo, Norway

**Keywords:** attention-deficit/hyperactivity disorder, ADHD, medication, typical development, children, executive functions, EF, attention, KiTAP

## Abstract

The emerging literature reports that children with Attention-Deficit/Hyperactivity Disorder (ADHD) show deficits in executive functioning. To date, the combination of drug therapy with certain evidence-based non-medication interventions has been proven to be the most effective treatment for ADHD. There is a gap in the literature regarding comparing the executive functions (EF) of treatment naïve and medicated children with ADHD with both each other and typically developing children. Altogether, 50 treatment naïve and 50 medicated children with ADHD and 50 typically developing children between the ages of six and 12 were enrolled. The Mini International Neuropsychiatric Interview for Children and Adolescents (Mini Kid) and the Test of Attentional Performance for Children (KiTAP) measures were employed. Treatment naïve children with ADHD showed weaker performance on most executive function measures (12 out of 15) than either the medicated ADHD group or the controls. There were no significant differences between the medicated ADHD children and typically developing children in most KiTAP parameters (10 out of 15). Executive function impairments were observable in treatment naïve ADHD children, which draws attention to the importance of treating ADHD. Future studies should focus on the specific effects of stimulant medication on executive functions.

## 1. Introduction

Attention-Deficit/Hyperactivity Disorder (ADHD) is common in childhood and adolescence with a prevalence of 4–6% [1,2]. It is included in the Diagnostic and Statistical Manual of Mental Disorders, 5^th^ edition (DSM-5) under “Neurodevelopmental Disorders” [3]. Although ADHD has been broadly studied in recent decades, many aspects of its etiology are still poorly understood. Researchers have suggested that the cause of the disorder is linked to the frontal regions of the brain (e.g., [4,5]). Several studies have reported structural [6,7,8] and functional [9,10,11] abnormalities in the brain networks [12], where abnormalities are suggested to be associated with impairment in the cognitive, affective, and motor behaviors observed in ADHD [13,14]. For example, a magnetic resonance imaging (MRI) study showed significantly smaller gray matter volume in adolescents with ADHD when compared to the control participants within the anterior cingulate cortex (ACC), the occipital cortex, bilateral hippocampus/amygdala, and also in extensive cerebellar regions [15]. Furthermore, significant cortical thinning was detected in the right rostral anterior cingulate cortex in children with ADHD when compared with the healthy controls [16]. Furthermore, reductions in the ACC gray matter volume was found to be related to deficits in attention in subjects with ADHD [15].

Researchers have argued that the hyperactive, impulsive, and inattentive behavior of children with ADHD is a consequence of various executive function (EF) deficits (e.g., [17,18,19,20,21]). Barkley [17] specifically indicated that ADHD is primarily characterized by a deficiency in behavioral inhibition, which results in further impairments in four executive functions: (1) non-verbal working memory; (2) internalization of speech (verbal working memory); (3) self-regulation of affect, motivation, and arousal; and (4) reconstitution. Barkley’s [17] theory is supported by the results of numerous empirical studies that demonstrate the involvement of inhibition [22,23,24,25], non-verbal and verbal working memory [22,23,24,26], the internalization of speech [27,28], self-regulation [29], and planning [22,23,26,30,31] in ADHD. In contrast, many researchers (e.g., [32,33,34,35,36,37]) have questioned the central role of executive functions in ADHD.

Aside from EFs, other cognitive skills are also thought to be impaired in ADHD. Various studies have shown impairment regarding alertness [38,39,40], an increased vulnerability to distraction [3,41,42,43], difficulties with divided attention [44], cognitive flexibility [24,34,35], and inhibition [45,46,47,48,49,50] in individuals with ADHD. In more detail, results of various neuropsychological studies displayed impairments in alertness in ADHD [39,40], and for distractibility [41,42], which is also a key feature of ADHD, one of its DSM-5 criteria [3]. In an effort to conduct ecologically more valid studies, Adams et al. [43] examined distractibility in individuals with ADHD in a virtual reality classroom using simulated ‘real-world’ auditory and visual distractors. Results indicated that distractors affected children with ADHD significantly more, resulting in worse performance than those children without ADHD. Divided attention has also been the focus of ADHD research [51,52], while other results have confirmed that deficits in cognitive flexibility may be present in ADHD [24,34,35]. However, some of these studies did not detect difficulties regarding divided attention [51,52] and cognitive flexibility [53]. Finally, moderate deficits in common paradigms measuring inhibitory control are typically observable in children with ADHD when compared to their typically developing peers (see [45,46,47,48,49,50]).

The combination of certain non-pharmacological [54,55,56] and medication treatments [54,57,58] has proven to be the most effective way of managing ADHD [59]. Some of the studies analyzing the effects of medication on cognitive functioning found positive effects of stimulant medication on reaction time (RT) variability [60], spatial short-term memory, spatial working memory, set-shifting and planning ability [61], attention, response inhibition, writing, and verbal working memory [62]. In a systematic review and meta-analysis by Coghill et al. [63] methylphenidate was found to be superior to the placebo in its effect of enhancing executive memory, non-executive memory, response inhibition, and in reducing reaction time and reaction time variability. However, one study found that individually tailored doses of methylphenidate did not reduce the cognitive impairments typically associated with ADHD [64].

Due to the contradicting results of previous findings and in order to gain more insight about performance on a broad range of cognitive tasks, the aim of our current study was to investigate whether children with ADHD could be distinguished by differences in their executive functioning and attentional performances from (1) typically developing children (children without ADHD) and (2) from children with ADHD who received adjusted medical treatment. Consequently, we hypothesized that treatment naïve ADHD children would perform significantly worse on each of the measured parameters (errors, omissions, reaction time, and variability of reaction time) of a widely used executive function and attention battery than either the medicated or the typically developing group. Furthermore, we presumed that there would be no significant differences in the measured parameters between the medicated and the typically developing group.

## 2. Materials and Methods

### 2.1. Participants

A total of 168 children aged between six and 12 were recruited to participate in the study between February 2016 and February 2018. The clinical sample was recruited in the Vadaskert Child Psychiatric Hospital and Outpatient Clinic, Budapest, Hungary. The control group consisted of typically developing children from elementary schools in Budapest, Hungary.

For the clinical sample, the inclusion criteria were that the children had to be between six and 12 years old with a diagnosis of ADHD. ADHD diagnosis was based on a structured diagnostic interview (see below). We specified two clinical samples: (1) treatment naïve children with ADHD and (2) children with ADHD receiving adjusted medical treatment.

The treatment naïve ADHD group contained 40 children (80%) with combined type, eight children (16%) with mostly inattentive type, and two children (4%) with mostly impulsive/hyperactive type of ADHD. Regarding the medicated sample, 48 children (96%) had the diagnosis of combined type, one child (2%) had the mostly inattentive type, and one child (2%) was diagnosed with the mostly impulsive/hyperactive type of ADHD. We included children with the diagnosis of ADHD based on a structured diagnostic interview (Mini Kid, see below) to the ADHD groups. This diagnostic interview measures psychiatric diagnoses according to the DSM-IV criteria. To examine the different ADHD subtypes between the clinical groups, three χ^2^ tests were conducted for the three subgroups. As for the mostly inattentive group, the Pearson’s Chi-squared test with Yates’ continuity correction showed barely significant results, (χ^2^(1) = 4.4, *p* = 0.04). The hyperactive-impulsive subtype did not differ at all according to the medication (χ^2^(1) = 0, *p* = 1). The combined diagnosis differed according to group (χ^2^(1) = 4.6, *p* = 0.03). The results were not unexpected considering that children with the most severe symptoms (combined ADHD diagnosis) are under medication, whereas the “less severe” cases (either inattentive or hyperactive) are not.

Children in the treatment naïve group were seeking professional help for the first time in their life at the Vadaskert Child Psychiatric Hospital and Outpatient Clinic, Budapest, Hungary and the ADHD diagnosis was established. We contacted these children and their parents at this time and gave them information about our research project. We had the opportunity to conduct the study examination processes before their regular doctor could possibly suggest medication for the treatment of ADHD.

Regarding the medicated group, 43 (86%) children received methylphenidate treatment, while seven (14%) used atomoxetine. The average dose of methylphenidate was 15.7 mg (SD = 7.68) and 39.3 mg (SD = 15.66) for the atomoxetine. All children from the medicated group took their adjusted medication on the day of KiTAP testing.

The inclusion criteria for the treatment naïve ADHD children was that the children had never been treated with either methylphenidate or atomoxetine, while the medicated clinical sample had to receive ongoing treatment with either methylphenidate or atomoxetine.

The inclusion criteria for the control group stated that participants must not have received a psychological, psychiatric, or neurological diagnosis or associated treatment during the course of the study or in their medical history. In addition, the structured diagnostic interview must have confirmed the absence of ADHD.

The exclusion criteria in all study groups were intellectual disabilities and autism spectrum disorder in the medical history. Further exclusion criterion for every group were reluctance to complete each task; illness (for example, diarrhea, stomach ache); the use of other medications (Tiapride, Risperidone); an incomplete diagnostic interview; retrospectively established autism or intellectual disabilities; and former psychological, psychiatric, or neurological treatment in the case of the control children. With one child, the diagnostic interview could not be completed as he simply refused to continue with it, he became impatient, and without giving any explanation, he left the room.

The study was approved by the Ethical Committee of the Medical Research Council, Hungary (ETT-TUKEB). The parents of each child included in this study provided written informed consent after having been informed of the nature of the study.

Ethics Approval and Consent to Participate: Ethical permission was submitted and accepted on 25 January 2016 by the ETT-TUKEB, permission number: 5677-1/2016/EKU [89/16]).

### 2.2. Measures

#### 2.2.1. Mini International Neuropsychiatric Interview for Children and Adolescents (Mini Kid)

Psychiatric disorders were diagnosed with a modified version of the Hungarian Mini Kid [65,66,67,68,69]. The Mini Kid is a short, structured diagnostic interview, assessing 25 child psychiatric disorders according to the Diagnostic and Statistical Manual of Mental Disorders, 4th Edition (DSM-IV), and the modified version also measures subthreshold disorders [70]. Sheehan et al. [69] examined the concurrent validity and reliability of the Mini Kid and all of the examined parameters gave acceptable results. Balázs et al. [65] developed the Hungarian version of the Mini Kid. The high quality of the Hungarian translation of the Mini Kid was assured by a multistep translation procedure and its reliability was found to be good to excellent, except in the case of test–retest reliability for generalized anxiety disorder (κ = 0.36). When the current study started, the Hungarian version of the DSM-5 based Mini Kid was not available yet. The last author of this study (J.B.) is the Hungarian developer of Mini Kid and by now, the Hungarian version of DSM-5 has been evaluated. In our study, children were interviewed with their parents, according to the instructions of the Mini Kid administration procedure [65,69]. The Mini Kid was completed by one of the authors (M.M.), who is a psychologist. She underwent Mini Kid training before the study and was continuously supervised during the study by the last author of this manuscript (J.B.), who is a child psychiatrist.

A structured parents-rated questionnaire was specifically developed for this study to explore the demographic characteristics of the participants such as the parents’ education and economic activity, family structure, children’s motor development, and former and current psychological, psychiatric, or neurological treatment.

#### 2.2.2. Test of Attentional Performance for Children (KiTAP)

The KiTAP is a computer-based continuous performance task (CPT) and executive function (EF) battery [71]. The KiTAP is the child version of the Tests of Attentional Performance (TAP), a test used since the late 1990s to measure attention and EF performance in adults with different medical, neurological, and psychiatric conditions. KiTAP was constructed by considering the same concept as the adult version [44]. It was modified to be child-friendly by fitting the tasks to an enchanted castle story. The reason for this was that children found the existing test batteries boring and repetitive, and there had been problems with motivating young children to complete such test procedures at the Clinic for Rehabilitation of Children and Adolescents in Gailingen [44]. KiTAP has been translated to and validated in the following languages: English, French, Spanish, and Italian from its original German [72] version. The battery contains eight subtests and in this study, we used five of them: alertness, distractibility, divided attention, flexibility, and reaction control (inhibition) [71].

Intrinsic alertness is assessed with a simple reaction task (The witch): a witch appears at a window and should be chased away as fast as possible by pressing the reaction key button [44].

The aim of the distractibility subtest (Happy and Sad Ghosts) is to complete a centrally presented decision task (type go/no-go), whilst in half of the trials, a distracting stimulus emerges in the periphery [44]. The central stimulus is either a cheerful or sad ghost and can be only differentiated by visually focusing on their mouthline. The distractor turns up shortly before the central stimulus (400 ms), thus one saccade to the distracting stimulus is possible before the discriminative stimulus occurs. The appearance of the of the central stimulus is short (200 ms) and it typically already disappears before fixation on it is possible. The consequences of switching the orientation of attention caused by the distractor are omissions of the crucial stimulus or false reactions.

The divided attention subtest (The Owl) is a dual task, requiring processing of an auditory stimulus in addition to visual stimuli, and request subjects to simultaneously listen to a series of high and low owl sounds as well as watch for target stimuli (the owl with closed eyes) [44]. Subjects must press a button either when a sound is repeated or when the target stimulus appears.

The flexibility subtest (The Dragon’s House) demands the subjects alternate between identifying blue and green dragons that appear on random sides of the screen by tapping one of two buttons (numbered 1 and 2) [44]. This task analyses the ability of fast adjustment to a new condition by recording the response to constantly changing target stimuli.

The go/no-go or inhibition subtest (The Bat) requires subjects to tap a button when the target stimulus (a bat) appears, while not hitting the button for the non-target stimulus (a cat) [44]. It measures the ability of control and decision making by requiring a reply for one stimulus as fast as possible, while requiring no reply for the other.

It took approximately 30 min to complete these five KiTAP subtests. The subtests were administered in a quasi-randomized order. During the tests, subjects were supported by a research assistant (RA). Before the start of each subtest, the RA explained the goal of the test. The subject then took a short pretest, allowing them to understand the task.

### 2.3. Statistical Analyses

Data analysis was performed using R (3.5.1 version, R Foundation for Statistical Computing, Vienna, Austria).

The measures from each KiTAP subtest were analyzed including errors, omissions, median reaction time, and variability (standard deviation) of reaction time. For continuous variables (median and variability of reaction time), we ran normality (Shapiro) and homogeneity (Bartlett) tests on the raw data. The Tukey power transformation was subsequently used if the normality assumption was violated. An ANOVA was mainly estimated on the Tukey transformed data. After estimating the ANOVA models, potential assumption violations were inspected by using the Global Validation of Linear Models Assumptions (GVLMA) package [73]. In the next step, all control variables were added. Control variables were the following: gender, age, mother’s education, father’s education, and the number of months spent doing sports. The level of parental education seems to affect the children’s cognition [74] and executive functioning [75]. Physical activity was found to have an effect on cognition among typically developing (e.g., [76,77,78,79]) and children with ADHD [80,81,82]. After adding the control variables, the best fitting model was selected with the stepAIC (it performs stepwise model selection by Akaike Information Criterion) function from the MASS (contains functions and datasets to support Venables and Ripley, “Modern Applied Statistics with S”, 4th edition, 2002) package [83]. This estimates all models obtained by both forward and backward selection and chooses the best model fit based on the lowest Akaike Information Criterion. In all models, we kept the group and age variables. On one hand, the former was our focal predictor. On the other hand, we also had to address the significant association between group membership and age, hence we controlled for the latter in all regressions. After this, the linear model’s assumptions were assessed. Post hoc tests were carried out with Tukey’s honest significant adjustment (HSD): both the groups were compared pairwise, and the non-medicated group was contrasted with the mean of the medicated and control groups.

Regarding count data (errors and omissions), we used the generalized linear model. The dispersion parameter was higher than the unity for all dependent variables, hence we preferred negative binomial regressions over Poisson regressions. Subsequently, multiple model specifications were tested. First, we compared the fit of the linear versus the quadratic variance functions (the overdispersion of the variance is modeled as a linear versus quadratic function of the mean). Second, due to the large number of zeros being observed in the dependent variable, we tested the presence of zero-inflation. Model fits were compared based on the likelihood ratio test, in addition to the Akaike and Bayesian Information Criterion. Subsequently, parameters were estimated for the group variable from the best-fitting model. This was followed by estimating the group effect and the three post hoc tests with Tukey’s HSD correction. Finally, we contrasted the non-medicated group against the mean of the other two groups.

After estimating all 15 models, we controlled for family-wise error to decrease the risk of a Type I error. We used the method provided by Benjaini and Yekutieli [84] for multiple reasons. On one hand, it is a conservative test, but it still does not reduce the power of the test. On the other hand, it takes into account the regression dependency between each of the test statistics corresponding to the true null hypothesis.

## 3. Results

### 3.1. Sample

This paper presents data for 50 treatment naïve children with ADHD (45 boys and five girls, age: M = 8.26 years, SD = 1.47, aged 6–11 years), a further 50 children with ADHD with ongoing adjusted medical treatment (47 boys and three girls, age: M = 9.70 years, SD = 1.78, aged 6–12 years), and 50 control group children (43 boys and seven girls, age: M = 8.68 years, SD = 1.41, aged 6–11 years). Generally, the sample consisted mainly of boys (135 boys versus 15 girls). Significant age differences were found between the groups (χ^2^(2) = 17.802, *p* < 0.001). Table 1 provides an overview of the participants’ characteristics.

### 3.2. KiTAP Parameters

In the following section, we will present the results of the different KiTAP subtests. In the univariate models, the ‘Group’ variable was always significant. We tested the presence of zero-inflation for all 15 final models, but none of them were significant. Table 2 shows the results of the post hoc tests for each examined variable (KiTAP parameters) and the effects of the covariates. Table 3 displays the descriptive statistics, whilst Table 4 contains the results of contrasting the non-medicated group to the medicated and control groups.

The first three columns contain the results of the post hoc test after estimating an ANCOVA. The first number shows the contrast estimate between the two groups followed by its standard error in parentheses. Subsequently, the corresponding t-values or z-values (depending on whether we estimated ANOVA or GLM) and *p*-values are shown. The last column demonstrates the significance of the added predictors, the F-value, or χ^2^ value (depending on whether the ANOVA or GLM model was estimated) of the group variable along with the significant covariates.

#### 3.2.1. Alertness

With reference to task alertness, the following two measures were of interest: the median of the reaction time and variability (standard deviation) of the reaction time. Regarding the median of the reaction time, out of the possible control variables, the age of the participants proved to be significant by using ANCOVA (F(1, 146) = 17.25, *p* < 0.001). Group membership had a significant effect (F(2, 146) = 4.66, *p* = 0.01; after controlling for family-wise error *p* = 0.04), however, the pairwise contrasts after Tukey correction did not show any significant differences (*p* > 0.05) (Table 2). Neither did contrasting the average of the medicated and control participants result in significant differences (Table 4).

Regarding the variability of reaction time (standard deviation of reaction time) among the variables, the age of the participants (F(1, 146) = 8.43, *p* = 0.004) proved to be significant using ANCOVA. Group membership was highly significant (F(2, 146) = 12.22, *p* < 0.001) and also remained so after the Benjaini and Yekutieli test (*p* < 0.001). The corrected contrast estimates between the non-medicated and control (t(146) = 3.66, *p* = 0.001), between the non-medicated and medicated (t(146) = 3.19, *p* = 0.005) children (Table 2) as well as the non-medicated group against the averaged performance of the other two groups (t(146) = −3.94, *p* < 0.001) proved to be significant (Table 4).

#### 3.2.2. Distractibility

A Generalized Linear Negative Binomial Model was used on the total number of omissions. Age (χ^2^(1) = 6.37, *p* = 0.01), mother’s education level (χ^2^(3) = 9.25, *p* = 0.03) were significant variables, and months spent with sports was marginally significant (χ^2^(1) = 2.98, *p* = 0.09). Group effect was also significant (χ^2^(2) = 14.23, *p* < 0.001, after the Benjaini and Yekutieli test *p* = 0.005). The corrected contrasts showed a significant difference between the non-medicated and control (z = 2.75, *p* = 0.02) as well as between the medicated and control (z = 2.51, *p* = 0.03) participants (Table 2). Contrasting the two groups against non-medicated children showed a marginal significance (z = −1.7, *p* = 0.09) (Table 4).

To examine the number of omissions made with a distractor, a generalized linear negative binomial model was conducted. Age (χ^2^(1) = 9.41, *p* = 0.002), mother’s education level (χ^2^(3) = 14.73, *p* = 0.002), and months spent with sport (χ^2^(1) = 1.73, *p* = 0.19) were the relevant control variables, and group reached significance (χ^2^(2) = 10.71, *p* = 0.005), even after correction for family-wise error (*p* = 0.02). Corrected group-wise contrasts indicated significant differences between non-medicated and control (z = 2.47, *p* = 0.04) as well as between medicated and control (z = 2.42, *p* = 0.04) (Table 2), while contrasting non-medicated children against the other two groups did not show significant differences (Table 4).

Additionally, a generalized linear negative binomial model was used on the number of omissions without a distractor. Only the group variable was significant (χ^2^(2) = 10.78, *p* = 0.005 even after the Benjaini and Yekutieli test *p* = 0.02), however, to control for the effect of age differences between the groups, we also added the age variable (χ^2^(1) = 0.89, *p* = 0.35). After applying the Tukey correction, only non-medicated and control groups differed significantly (z = 3.03, *p* = 0.007), and medicated and control groups showed a marginally significant difference (z = 2.07, *p* = 0.1) (Table 2). Non-medicated children had significantly lower performance than the average of the other two groups (z = −2.92, *p* = 0.02) (Table 4).

A generalized linear negative binomial model was conducted on the total number of false reactions. Age (χ^2^(1) = 38.7, *p* < 0.001) and months spent with sports (χ^2^(1) = 5.7, *p* = 0.02) were relevant covariates, and group effect was highly significant (χ^2^(2) = 18.3, *p* < 0.001), even after family-wise error correction (*p* < 0.001). Non-medicated and medicated groups did not differ, however, both the former (z = 3.04, *p* = 0.007) and the latter (z = 2.46, *p* = 0.04) differed from the control participants (Table 2). The mean of controls and medicated patients showed a significant contrast against non-medicated children (z = −2, *p* = 0.05) (Table 4).

Focusing on the number of errors with a distractor, a generalized linear negative binomial model was conducted including age (χ^2^(1) = 31.9, *p* < 0.001) and months spent doing sports (χ^2^(1) = 8, *p* = 0.005). Group effect was highly significant (χ^2^(2) = 21.5, *p* < 0.001, after the Benjaini and Yekutieli test *p* < 0.001). After the Tukey correction, only non-medicated and control participants were proven to perform differently (z = 3.2, *p* = 0.004) (Table 2), however, when grouping together the medicated and control participants, they had a significant contrast with the non-medicated children (z = −2.59, *p* = 0.01) (Table 4).

Finally, a generalized linear negative binomial model was also used on the number of errors without a distractor. Age (χ^2^(1) = 35.2, *p* < 0.001) and months spent with sports (χ^2^(1) = 2.2, *p* = 0.14) were relevant controls, and group effect was significant (χ^2^(2) = 12.6, *p* = 0.002), even after the correction for family-wise error (*p* = 0.01). Non-medicated and medicated groups showed similar performance, but both the first group (z = 2.59, *p* = 0.03) and the second group (z = 2.68, *p* = 0.02) differed from the control participants (Table 2). Contrasting the non-medicated group against the other two did not show relevant differences (Table 4).

#### 3.2.3. Divided Attention

A generalized linear negative binomial model was conducted on the total number of omissions (number of missed signals). Among the variables, the age of the participants (χ^2^(1) = 14.1, *p* < 0.001) and group effect (χ^2^(2) = 45.6, *p* < 0.001, after family-wise error correction *p* < 0.001) proved to be significant. After the Tukey correction, a significant difference was found between the medicated and non-medicated (z = 3.56, *p* = 0.001), and also between the non-medicated and control (z = 6, *p* < 0.001) groups (Table 2). Non-medicated children performed significantly different than the average of the medicated and control group children (z = −5.66, *p* < 0.001) (Table 4).

A generalized linear negative binomial model was conducted on the total number of false reactions (errors). Age (χ^2^(1) = 19.5, *p* < 0.001), father’s education level (χ^2^(3) = 9.4, *p* = 0.03), and months spent doing sports (χ^2^(1) = 5.7, *p* = 0.02) were significant covariates. Group effect was found to be highly significant (χ^2^(2) = 52, *p* < 0.001), even after the Benjaini and Yekutieli test (*p* < 0.001). After applying Tukey correction, significant differences were yielded between the medicated and non-medicated children (z = 3.4, *p* = 0.002), and between the non-medicated and control children (z = 4.82, *p* < 0.001) (Table 2). There was no significant difference between the medicated and control group. Comparison between the average of the non-medicated group versus the average of the medicated and control group indicated a significant difference (z = −4.86, *p* < 0.001) (Table 4).

Regarding the median reaction time, age (F(1, 146) = 41.7, *p* < 0.001) and group effect (F(2, 146) = 13.3, *p* < 0.001, after family-wise error correction *p* < 0.001) were significant by using ANCOVA on the raw data of the median of reaction time. Corrected group-wise contrasts resulted in a significant difference for the non-medicated group when compared with the medicated (t(146) = 2.51, *p* = 0.01) and control group (t(146) = 2.48, *p* = 0.01) (Table 2). The non-medicated group against the averaged performance of the other two groups proved to be significant (t(146) = −2.52, *p* = 0.01) (Table 4).

#### 3.2.4. Flexibility

A generalized linear negative binomial model was used for false reactions. Among the variables, age (χ^2^(1) = 23.2, *p* < 0.001) and group effect (χ^2^(2) = 24.8, *p* < 0.001, after the Benjaini and Yekutieli test *p* < 0.001) proved to be significant. Only the difference between the non-medicated and control group (z = 3.95, *p* < 0.001) was found to be significant (Table 2). A marginally significant difference was yielded between the non-medicated and medicated children (z = 2.26, *p* = 0.06). The comparison between the medicated and control group did not result in a significant difference. Additionally, when grouping together the medicated and control children, it gave a significant contrast with the non-medicated children (z = −3.64, *p* < 0.001) (Table 4).

Using transformed data of the median reaction time, the results of Bartlett’s test of homogeneity of variances were significant (Bartlett’s K-squared = 10.6, df = 2, *p* = 0.005). Among the variables that were applied, the age of the participants (F(1, 146) = 46.11, *p* < 0.001) and the group effect (F(2, 146) = 5.39, *p* = 0.006, after family-wise error correction *p* = 0.02) proved to be significant by ANCOVA. However, when including the age variable, we removed the heteroscedasticity in the error term, hence we used regular standard errors. The only significant difference occurred between the medicated and control group (t(146) = 2.37, *p* = 0.05). A marginally significant difference was detected between the non-medicated and control group (t(146) = 2.33, *p* = 0.06), whereas the difference was not significant between the medicated and non-medicated group (Table 2). Contrasting the non-medicated group against the medicated and control group did not result in relevant differences (Table 4).

#### 3.2.5. Go/No-Go

A generalized linear negative binomial model was used on the false reactions data. Among the variables, the age of the participants (χ^2^(1) = 14, *p* < 0.001) and group effect (χ^2^(2) = 21.9, *p* < 0.001, after correction for family-wise error *p* < 0.001) were found to be significant. After applying the correction, significant differences were yielded between the non-medicated and medicated (z = 2.43, *p* = 0.04), and also between the non-medicated and control group (z = 3.61, *p* < 0.001) (Table 2). The difference was not significant between the medicated and control children. The comparison between the average of the non-medicated children and the average of both the other groups revealed a significant difference (z = −3.56, *p* < 0.001) (Table 4).

Among the control variables, the age of the participants (F(1, 146) = 23.04, *p* < 0.001) and the effect of the group (F(2, 146) = 4.57, *p* = 0.01, after the Benjaini and Yekutieli test *p* = 0.04) proved to be significant using ANCOVA on the transformed data of the median of reaction time. Although this was the last significant group effect, the group-wise contrasts after Tukey correction did not display any significant differences (*p* > 0.05) (Table 2). It did not result in significant differences either, when contrasting the average of the medicated and control participants with the average of the non-medicated children (Table 4).

## 4. Discussion

The present study set out to investigate whether (1) the presence of ADHD symptoms was linked to impairment in EF compared to typically developing children, and (2) whether medication attenuated this association. The variables with the strongest relationship to the EF measures were the diagnosis of ADHD, receiving medication, age, physical activity, and parents’ education level.

### 4.1. Alertness: Reaction Time and ADHD

Regarding the median of reaction time, the effect of the group was significant; however, the post hoc tests revealed no significant differences among the three groups (ADHD-medicated, ADHD-non-medicated, and control). Descriptive parameters displayed considerably higher means and standard deviations of median reaction time for the group of treatment naïve children with ADHD than both the medicated and control groups. In contrast with this nonsignificant result, Cao et al. [39] found deficits in alerting functions in children with ADHD, where deficits were associated with abnormal activities in the frontal and parietal regions. Similarly, Bolfer et al. [85] found that children with ADHD showed reaction times higher than typically developing controls, while Coghill et al.’s [63] study displayed the effect of methylphenidate over the placebo in reducing reaction time. Somewhat in line with our results, few effects of methylphenidate were found on reaction time speed [60] and its use did not affect the attention skills in children with ADHD [64].

Regarding the variability (standard deviation) of reaction time, our models showed significant differences between the non-medicated and control, and between the non-medicated and medicated children. However, there was no significant difference between the medicated and control group children. In line with our results, the literature points out that children with ADHD exhibit increased reaction time variability relative to nonclinical groups [86], and that medication has positive effects on the variability of reaction time [60]. Accordingly, there is strong evidence that stimulant treatment reduces reaction time variability during a range of cognitive tasks [63,86,87].

In the inattention symptoms from the DSM-5 [3], several items specify poor sustained attention, suggesting that impairments in sustained attention is an essential clinical aspect for ADHD. The concept of inattention is not expressed in cognitive terms in the DSM-5 [3], nevertheless neuropsychological studies could detect abnormalities in basic attentional processes regarding ADHD [88,89]. The intensity of attention includes alertness and sustained attention, which might be necessary for attentional selectivity [90,91]. Based on the previous research, we considered using the KiTAP task testing sustained attention, but the duration of this test was much longer than the test for alertness, and children were not willing to complete the 10-min-long task. Therefore, we chose the alertness task instead.

### 4.2. Distractibility: Omissions and Errors in ADHD

A low degree of distractibility is an important prerequisite of concentrated work and therefore is important for academic achievement [44]. Regarding omissions only, there was a high proportion of zeros across all three groups, however, the statistical test did not detect zero-inflation. Treatment naïve ADHD children made the most omissions and errors in both the presence and absence of distractors. Medicated ADHD children made less omissions and errors, and the least omissions and errors were made by the control group. The difference was always significant between the non-medicated ADHD and control group, but was never significant between the non-medicated and medicated group. Regarding the total number of omissions, these results implicate that ADHD children without medication perform significantly worse than typically developing children. Additionally, medication does not seem to help them perform “better” (making less omissions), hence resulting in significant differences with the typically developing children. The same outcome was found concerning the omissions made when distractors were presented, implicating less help from medication to focus when the distractor appeared. Regarding the number of total errors and errors without distractors, a significant difference was yielded between the medicated and control group. Medication did not seem to help regarding the aforementioned parameters, indicating significantly more mistakes, and interestingly, more errors when the distractor was not presented. However, no significant difference was found between the medicated and non-medicated, and between the medicated and control group as for the errors with the distractor. Furthermore, by all error types (total, with and without distractor), a significant difference occurred between the non-medicated and control group, but not between the non-medicated and medicated group. Partly supporting our findings, previous studies have shown that psychostimulant medication alters performance on EF tests in children with ADHD [61,92,93]. As a key feature, in line with the DSM-5 criteria [3], a higher level of distractibility could be detected by our treatment naïve group when compared to typically developing children. Elevated levels of distractibility in ADHD could result from several elements, alone or in combination, like failing to maintain focus on a task, higher degree of orientating to novel stimulus [94], and deficits in inhibition about incoming sensory stimuli [95]. Van Mourik et al. [94] used novel sounds and standard sounds while recording event-related potentials while children with and without ADHD completed a visual two-choice reaction time task. The results of this study were contradictory to ours, since despite the elevated orienting response to novel sound, the distraction could facilitate the performance of ADHD children temporarily, perhaps by increasing arousal to an optimal level, as it was indicated by the reduced omission rate. The results of this study implies that the presence of distractors might not always potentially have a beneficial effect on children with ADHD [94].

### 4.3. Divided Attention: Omissions, Errors, and Reaction Time in ADHD

In everyday life, it is natural to pay attention to a number of events and things simultaneously [44]. For omissions and errors, there was a high number of values close to zero, but zero-inflation did not occur. Treatment naïve ADHD children made the most omissions and errors, followed by medicated ADHD children. Control group children made the least omissions and errors. Significant differences were found in both cases between the non-medicated and medicated group and between the non-medicated and control group. In our sample, no significant differences arose between the medicated ADHD and control group. These outcomes support our expectations, namely treatment naïve ADHD children perform significantly worse by making more omissions and errors than medicated and control group children. These results contrast Lajoie et al.’s [96] findings, in which the authors found that comparisons between on–off medication conditions generally disclosed few differences as for sustained and selective attention measures and simple processing speed. On the contrary, concerning higher-level attention domains including shifting and divided attention, children on medication demonstrated a speed–accuracy trade-off, exhibiting greater accuracy, but slower completion times. When the ADHD group was compared to the controls, ADHD children in the medication condition were more accurate across all attention domains on all measures. Our results are also in contrast with the study of Elosúa, Del Olmo, and Contreras [51], where children with ADHD displayed less impairments when completing a test measuring divided attention than the participants in the typically developing group.

Regarding the median of reaction time, the medicated group exhibited the best performance. Based on the descriptive statistics, treatment naïve ADHD children had the highest median of reaction time, followed by the control children, but with the medicated group showing the best performance. Among these, two differences reached significance: between the non-medicated and medicated, and between the non-medicated and control group. A possible explanation for the results of the median reaction time might be that medication did indeed affect reaction time, resulting in significantly shorter reaction times for medicated ADHD children than non-medicated ones. Furthermore, the fact that control participants were largely examined in the afternoons could have resulted in a slightly longer mean of the median reaction time observed, possibly due to their tiredness. In a study by [97], 35 children with ADHD (between the age of nine and 12) and 35 healthy control children were examined. Contrary to our findings, comparisons with the control group showed that ADHD children reacted significantly faster on all of the measured attention tests including a divided attention test, where they also had fewer errors.

A proper functioning of divided attention is essential in most everyday situations, since one needs to process multiple information coming from multiple sources and often multiple modalities simultaneously. The following DSM-5 [3] ADHD criteria could be related to a malfunctioning of divided attention: often losing personal belongings and appearing not to pay attention when someone talks directly to the person.

### 4.4. Flexibility: Errors and Reaction Time in ADHD

During the flexibility task, non-medicated children made the most errors, followed by medicated children. Control children made the least number of errors in the flexibility task. However, only the difference between the non-medicated and the control groups reached significance in the post hoc tests. In a study, Etchepareborda and Mulas [98] investigated a group of 50 children diagnosed with ADHD (8 to 21 years old) and 50 typically developing children. Their attentional functions, inhibitory control mechanisms, and cognitive flexibility of the subjects were tested. At least 38% of the patients in the study displayed impaired cognitive flexibility. Patients with poor cognitive flexibility also had difficulties with attentional discrimination, as well as impulse and interference control. The authors noticed that the group that was characterized by cognitive rigidity, in addition to attentional disorder, could correspond to a complex subtype not sensitive to stimulants. In line with our outcomes, cognitive flexibility was found to correlate significantly and negatively with the level of hyperactivity/inattention, although in typically developing children [99]. In another study stimulant medication had a positive effect on cognitive flexibility among other measured cognitive abilities [61].

With regards to the mean of the median reaction times, non-medicated children were the slowest followed by the medicated ones. Children in the control group were the fastest. Only the difference between the medicated and the control children reached significance on the post hoc tests, indicating that the reaction time performance of the medicated children did not reach that of the control group. Although treatment naïve children had the greatest mean of median reaction time, they had the smallest standard deviation of this parameter.

Summing up these results, this task is likely to have been challenging for the medicated ADHD children; even though they made the same number of errors as the control group, they took significantly longer to respond. As demonstrated earlier, few effects have been measured on reaction time speed across multiple cognitive tasks, when using methylphenidate [60].

Flexibility is also an essential aspect of attentional performance [44]. It promotes controlling and redirecting the focus of attention. It manifests through a large group of activities such as adapting our attitude in an environment that frequently produces changes in order to reach our goals, or when alteration is needed regarding our goal, because the chosen activity does not lead to the destined outcome. Consequently, any difficulties in flexibility can potentially result in a substantial deterioration in everyday performance [44]. Therefore, understandably, several results indicate and confirm that deficits in cognitive flexibility are possible indicators of ADHD [24,34,35].

### 4.5. Go/No-Go: Errors and Reaction Time

Control processes contain our ability to control our reactions and behavior [44]. A considerable proportion of children made either zero or one error, but it did not lead to zero-inflation models. The results met our previous expectations, namely that the mean of errors was almost identical for the control and medicated groups, while unmedicated children with ADHD made significantly more errors than both the medicated and the control groups. Supporting our results, Koschack et al. [97], found that ADHD subjects made significantly more errors on a go/no-go task than their typically developing peers. In line with our findings, Wodka et al. [100] found a significant effect of diagnosis on errors of commission in go/no-go tests. That is, children with ADHD made significantly more errors than the controls. In children with ADHD, response inhibition appears to be a primary deficit that is observable even when the EF demand of the task is minimal. Although increasing working memory demand appears to obstruct response inhibition, this influence is similar in ADHD and typically developing children [100]. Several studies support that there is an impairment in inhibition in children with ADHD compared to their typically developing peers [50,51,52,53,54,55] and that methylphenidate treatment improves inhibitory control [85] or response inhibition [62,63].

Concerning reaction time, the control group reached a greater mean of the median reaction time when compared with the other two ADHD groups, but without significance. Non-medicated ADHD children had the greatest standard deviation of median reaction time, but medicated ADHD children ranked higher than the control group in this measure. In line with these results, we found no significant difference between the groups for the median reaction time in the post hoc tests, although the group effect was significant. In contrast, Koschack et al.’s [97] results revealed that reaction times of children with ADHD were faster on all attentional tasks when compared to typically developing children. Furthermore, also contrary to our results, Epstein et al.’s [60] study demonstrated limited effects of methylphenidate on reaction time speed, and implies that stimulant medication primarily influences reaction time variability during multiple cognitive tasks.

Impulsive behavior is also one of the distinctive aspects of the different types of ADHD (primary inattentive, primary hyperactive/impulsive, and combined type) in accordance with DSM-5 [3] and can be regularly observed in children with behavioral impairments. Symptoms describing hypermotility (e.g., fidgeting, running around, climbing on everything, etc.) or those defining interruption and difficulties regarding waiting also refer to impairments in inhibition. As Barkley [17] stated earlier, behavioral inhibition is a key feature of ADHD.

## 5. Limitations

The results of this study should be interpreted in the context of its limitations. First, there were significant differences in the ages and hometowns of the different groups, which could be a possible confounding factor in our study.

The next limitation is that the control children were mainly examined in the afternoon due to the fact that they could not be absent from school, while members of the clinical samples were available in the morning.

Another limitation is that although we excluded those children from the control group who had the diagnoses of autism spectrum disorder and intellectual disability and/or psychological, psychiatric, or neurological treatment in their medical history, and a further exclusion criterion in the control group was if the structural diagnostic interview could establish the diagnosis of ADHD, however, we did not exclude those children from any study groups who met the criteria or subthreshold criteria for any other psychiatric disorders after completing the Mini Kid.

Children from all study groups were excluded if intellectual disability was stated in their medical history, however, during the research process we did not test the level of intelligence.

The next limitation is about symptom severity. All children from the medicated group were receiving adjusted medication and were under this type of treatment at the time of the study. Children were taking either methylphenidate or atomoxetine. Therefore, symptom severity could not be assessed without medication because it would have raised even more ethical questions as for withdrawing atomoxetine treatment just to conduct the study.

The next limitation includes a significant difference between the medicated and non-medicated children regarding the combined ADHD subtype.

Finally, although our study groups included 50 participants each, so that we had enough power in the study, it would be necessary to also replicate our results in bigger sample size.

## 6. Conclusions

In summary, we would like to highlight that non-medicated ADHD children generally performed worse at different EF tasks than their typically developing peers. The means of the medicated group was closer to the means of the control groups rather than to the treatment naïve group in more cases, with the exception of the omissions made without a distractor in the distractibility test, and of the median reaction time in the go/no-go task. Medication seems to have had the strongest measurable effect on the children’s performance in the go/no-go (inhibition), the alertness, and the divided attention tasks. Treatment naïve ADHD children performed worse than the medicated and control group in almost all parameters, suggesting that ADHD is characterized by impaired EF. Future studies should focus on the effect of various medications on the EF in different ADHD subtypes.

## Figures and Tables

**Table 1 ijerph-16-03822-t001:** Sample characteristics.

Variables	Non-Medicated Group	Medicated Group	Control Group
Age (mean)	8.26	9.7	8.68
SD	1.47	1.78	1.41
Gender	45 boys and 5 girls	47 boys and 3 girls	43 boys and 7 girls
Residence (person)			
Capital	27	21	45
Countryside city	11	23	3
Village	12	6	1
Countryside town	0	0	1
Accommodation (person)			
Own parents	46	47	50
Adopted	3	3	0
Foster parents	1	0	0
Father education	7	9	1
Elementary	25	25	14
Intermediate	16	13	35
Higher	2	3	0
Mother education			
Elementary	8	3	0
Intermediate	15	24	11
Higher	26	22	39
N/A	1	1	0
Months spent with sports (mean)	24.4	33.2	35.3
SD	23.1	32	27.1

N/A: Not available; SD: Standard deviation.

**Table 2 ijerph-16-03822-t002:** Post hoc tests and covariates.

KiTAP Variables	Non-Medicated-Medicated	Non-Medicated-Control	Medicated-Control	Predictors (Significance and R^2^ Change)
Alertness				
Reaction time (RT) median				
Estimate:	0	0	0	Group: F(2, 146) = 4.66,
Standard error (SE):	0	0	0	*p* < **0.05**
t-value:	t(146) = 1.37	t(146) = 1.20	t(146) = −0.25	Age: F(1, 146) = 17.25,
*p*-value:	*p* > 0.05	*p* > 0.05	*p* > 0.05	*p* < **0.001**
RT variability				
Estimate:	0.01	0.01	0	Group: F(2, 146) = 12.22,
SE:	0	0	0	*p* < **0.001**
t-value:	t(146) = 3.19	t(146) = 3.66	t(146) = 0.26	Age: F(1, 146) = 8.43,
*p*-value:	*p* < **0.01**	*p* < **0.01**	*p* > 0.05	*p* < **0.01**
Distractibility				
Total omissions				
Estimate:	0.03	0.72	0.69	Group: χ^2^(2) = 14.23,
SE:	0.26	0.26	0.27	*p* < **0.001**
z-value:	z = 0.13	z = 2.75	z = 2.51	Age: χ^2^(1) = 6.37,
*p*-value:	*p* > 0.05	*p* < **0.05**	*p* < **0.05**	*p* < **0.05**
				Months spent with sport: χ^2^(1) = 2.98, *p* > **0.05**
				Mother education level: χ^2^(1) = 9.25, *p* < 0.05
Omissions with distractor				
Estimate:	−0.03	0.67	0.7	Group: χ^2^(2) = 10.71,
SE:	0.26	0.27	0.29	*p* < **0.01**
z-value:	z = −0.1	z = 2.47	z = 2.42	Age: χ^2^(1) = 9.41,
*p*-value:	*p* > 0.05	*p* < **0.05**	*p* < **0.05**	*p* < **0.01**
				Months spent with sport: χ^2^(1) = 1.73, *p* > 0.05
				Mother’s education level: χ^2^(3) = 14.73, *p* < **0.01**
Omissions without distractor				
Estimate:	0.32	1.05	0.73	Group: χ^2^(2) = 10.78,
SE:	0.31	0.35	0.35	*p* < **0.01**
z-value:	z = 1.03	z = 3.03	z = 2.07	Age: χ^2^(1) = 0.89,
*p*-value:	*p* > 0.05	*p* < **0.01**	*p* > 0.05 (marginally)	*p* > 0.05
				Months spent with sport: χ^2^(1) = 1.99, *p* > 0.05
Errors (total)				
Estimate:	0.05	0.33	0.28	Group: χ^2^(2) = 18.3,
SE:	0.11	0.11	0.11	*p* < **0.001**
z-value:	z = 0.46	z = 3.04	z = 2.46	Age: χ^2^(1) = 38.7,
*p*-value:	*p* > 0.05	*p* < **0.01**	*p* < **0.05**	*p* < **0.001**
				Months spent with sport: χ^2^(1) = 5.7, *p* < **0.05**
Errors with distractor				
Estimate:	0.14	0.35	0.21	Group: χ^2^(2) = 21.5,
SE:	0.11	0.11	0.12	*p* < **0.001**
z-value:	z = 1.25	z = 3.2	z = 1.79	Age: χ^2^(1) = 31.9,
*p*-value:	*p* > 0.05	*p* < **0.01**	*p* > 0.05	*p* < **0.001**
				Months spent with sport: χ^2^(1) = 8, *p* < **0.01**
Errors without distractor				
Estimate:	−0.03	0.32	0.35	Group: χ^2^(2) = 12.6,
SE:	0.13	0.13	0.13	*p* < **0.01**
z-value:	z = −0.21	z = 2.59	z = 2.68	Age: χ^2^(1) = 35.2,
*p*-value:	*p* > 0.05	*p* < **0.05**	*p* < **0.05**	*p* < **0.001**
				Month spent with sport: χ^2^(1) = 2.2, *p* > 0.05
Divided attention				
Omissions (total)				
Estimate:	0.74	1.23	0.49	Group: χ^2^(2) = 45.6,
SE:	0.21	0.21	0.22	*p* < **0.001**
z-value:	z = 3.56	z = 6	z = 2.17	Age: χ^2^(1) = 14.1,
*p*-value:	*p* < **0.01**	*p* < **0.001**	*p* > 0.05	*p* < **0.001**
Errors (total)				
Estimate:	0.63	0.91	0.28	Group: χ^2^(2) = 52,
SE:	0.19	0.19	0.2	*p* < **0.001**
z-value:	z = 3.4	z = 4.82	z = 1.38	Age: χ^2^(1) = 19.5,
*p*-value:	*p* < **0.01**	*p* < **0.001**	*p* > 0.05	*p* < **0.001**
				Months spent with sport: χ^2^(1) = 5.7, *p* < **0.05**
				Father’s education level: χ^2^(3) = 9.4, *p* < **0.05**
RT median				
Estimate:	65.8	49	−16.8	Group: F(2, 146) = 13.3,
SE:	21.5	19.7	20.8	*p* < **0.001**
t-value:	t(146) = 2.51	t(146) = 2.48	t(146) = −0.81	Age: F(1, 146) = 41.7,
*p*-value:	*p* < **0.05**	*p* < **0.05**	*p* > 0.05	*p* < **0.001**
Flexibility				
Errors (total)				
Estimate:	0.32	0.54	0.22	Group: χ^2^(2) = 24.8,
SE:	0.14	0.14	0.15	*p* < **0.001**
z-value:	z = 2.26	z = 3.95	z = 1.46	Age: χ^2^(1) = 23.2,
*p*-value:	*p* > 0.05 (marginally)	*p* < **0.001**	*p* > 0.05	*p* < **0.001**
RT median				
Estimate:	−0	0	0	Group: F(2, 146) = 5.39,
SE:	0	0	0	*p* < **0.01**
t-value:	t(146) = −0.1	t(146) = 2.33	t(146) = 2.37	Age: F(1, 146) = 46.11
*p*-value:	*p* > 0.05	*p* > 0.05 (marginally)	*p* < **0.05**	*p* < **0.001**
Go/no-go				
Errors (total)				
Estimate:	0.48	0.68	0.2	Group: χ^2^(2) = 21.9,
SE:	0.2	0.19	0.21	*p* < **0.001**
z-value:	z = 2.43	z = 3.61	z = 0.96	Age: χ^2^(1) = 14,
*p*-value:	*p* < **0.05**	*p* < **0.001**	*p* > 0.05	*p* < **0.001**
RT median				
Estimate:	0	−0	−0	Group: F(2, 146) = 4.57,
SE:	0	0	0	*p* < **0.05**
t-value:	t(146) = 0.36	t(146) = −1.2	t(146) = −1.54	Age: F(1, 146) = 23.04,
*p*-value:	*p* > 0.05	*p* > 0.05	*p* > 0.05	*p* < **0.001**

*p* values under 0.05 are shown in bold.

**Table 3 ijerph-16-03822-t003:** Descriptive statistics.

KiTAP Variables	Non-Medicated	Medicated	Control
Alertness			
RT median			
Mean	355.89	315.8	331.47
Median	342	306	326
Standard Deviation	82.01	57.36	62.29
Minimum	244.5	194	240
Maximum	602.5	521	491.5
RT variability			
Mean	125.02	62.76	62.9
Median	86.71	57.54	60.84
Standard Deviation	133.94	29.23	24.9
Minimum	32.47	25.85	27.72
Maximum	633.06	156.11	165.63
Distractibility			
Omission			
total			
Mean	3.46	2.4	1.38
Median	2	1	0.5
Standard Deviation	3.47	3.96	2.07
Minimum	0	0	0
Maximum	17	18	10
with distractor			
Mean	2.02	1.38	0.9
Median	1	1	0
Standard Deviation	2.13	2.28	1.5
Minimum	0	0	0
Maximum	9	10	8
without distractor			
Mean	1.44	1.02	0.48
Median	1	0	0
Standard Deviation	1.96	1.89	1.01
Minimum	0	0	0
Maximum	11	8	5
Error			
total			
Mean	18.54	13.84	11.88
Median	19	11.5	10.5
Standard Deviation	9.45	8.58	7.24
Minimum	1	1	1
Maximum	39	31	29
with distractor			
Mean	9.24	6.44	5.84
Median	9	6	5.5
Standard Deviation	4.61	4.17	3.51
Minimum	1	0	0
Maximum	18	16	12
without distractor			
Mean	9.3	7.4	6.04
Median	9	6	5
Standard Deviation	5.3	4.97	4.34
Minimum	0	0	0
Maximum	21	18	17
Divided attention			
Omission			
Mean	6.24	2.46	1.8
Median	6	1	1
Standard Deviation	4.53	3.49	2.07
Minimum	0	0	0
Maximum	17	21	10
Error			
Mean	22	9.66	6.64
Median	14	7	5.5
Standard Deviation	22.67	11.97	5.46
Minimum	0	0	0
Maximum	118	68	20
RT median			
Mean	813.66	710.24	762.05
Median	825	693	750
Standard Deviation	116.44	119.23	104.14
Minimum	492.5	493.5	599
Maximum	1121	959.5	1117
Flexibility			
Error			
Mean	5.92	3.42	3.34
Median	6	3	3
Standard Deviation	3.28	3	2.59
Minimum	0	0	0
Maximum	14	12	12
RT median			
Mean	1069.88	999.96	930.64
Median	1012	911.5	889
Standard Deviation	246.71	413.49	251.04
Minimum	645	456	528
Maximum	1929.5	2804.5	1643
Go/No-Go			
Error			
Mean	4.28	2.12	2.04
Median	3	1	1
Standard Deviation	3.72	2.25	2.08
Minimum	0	0	0
Maximum	19	9	7
RT median			
Mean	507.63	474.2	513.14
Median	490.5	465.75	515.25
Standard Deviation	91.65	73.31	69.38
Minimum	349.5	351	398
Maximum	734	668	690.5

**Table 4 ijerph-16-03822-t004:** Contrasting non-medicated group with the mean of the other two groups.

KiTAP Variables	Estimate	Significance
Alertness		
RT median	−0	t(146) = −1.49, *p* > 0.05
RT variability	−0.01	t(146) = −3.94, *p* < **0.001**
Distractibility		
Total omission	−0.38	z = −1.7, *p* > 0.05
Omission with distractor	−0.32	z = −1.44, *p* > 0.05
Omission without distractor	−0.65	z = −2.29, *p* < **0.05**
Total error	−0.19	z = −2, *p* < **0.05**
Error with distractor	−0.24	z = −2.59, *p* < **0.05**
Error without distractor	−0.15	z = −1.37, *p* > 0.05
Divided Attention		
Total omission	−0.99	z = −5.66, *p* < **0.001**
Total error	−0.77	z = −4.86, *p* < **0.001**
RT median (raw data)	−45.6	t(146) = −2.52, *p* < **0.05**
Flexibility		
Total error	−0.43	z = −3.64, *p* < **0.001**
RT median	−0	t(146) = −1.24, *p* > 0.05
Go/no-go		
Total error	−0.58	z = −3.56, *p* < **0.001**
RT median	0	t(146) = 0.46, *p* > 0.05

*p* values under 0.05 are shown in bold.

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
