# Peer review of "Executive Function and Attention Performance in Children with ADHD: Effects of Medication and Comparison with Typically Developing Children"

_ijerph, 2019, doi:10.3390/ijerph16203822_

Round 1

Reviewer 1 Report

  I appreciate the opportunity to review the manuscript entitled “Executive function in children with ADHD: effects of medication and comparison with typically developing children.” The current study compares three groups of children (children receiving ongoing stimulant and non-stimulant treatments for ADHD, medication naïve children with ADHD, and typically developing children) on measures of executive functioning using the KiTAP battery. The topic is important and timely to the field as ADHD is increasingly recognized as a neurodevelopmental disorder with significant impairments in a variety of executive functions including, but not limited to, working memory, behavioral inhibition, planning/organization, as well as increased reaction time variability. Additionally, while a number of studies have investigated the effects of psychopharmacological treatments on executive functions in children with ADHD, few directly compare medication naïve children with ADHD to medicated children with ADHD and typically developing children within the same study. My main concerns include (1) an incomplete and outdated literature review in both the introduction and discussion sections, (2) missing information regarding medication dosing, medication type, and whether or not they were medicated (for the medication group) during the time of the KiTAP testing appointment, and (3) statistical concerns and confusing reporting of results. The following comments are subdivided by subsections within the manuscript and are recommended to strengthen the clarity of the study and readability of the paper.

Title:

I think the title is a bit misleading as it suggests that the study is a clinical trial, but it is instead comparing a group of medication naïve ADHD children to medicated ADHD children and typically developing children. It would be particularly misleading if children were not required (or were not recorded as) taking their medication the day of the KiTAP testing.

Introduction:

In the abstract, the authors state a gap in the research literature regarding EFs of treatment naïve and medicated children with ADHD compared to TD children, but do not review any of this literature in the introduction. I found this particularly surprising, as there have been a number of studies, including a meta-analysis, examining medication effects on cognitive functioning in children with ADHD. See the following studies listed below:

Coghill, D. R., Seth, S., Pedroso, S., Usala, T., Currie, J., & Gagliano, A. (2014). Effects of methylphenidate on cognitive functions in children and adolescents with attention-deficit/hyperactivity disorder: evidence from a systematic review and a meta-analysis. Biological psychiatry76(8), 603-615.

Epstein, J. N., Brinkman, W. B., Froehlich, T., Langberg, J. M., Narad, M. E., Antonini, T. N., … Altaye, M. (2011). Effects of stimulant medication, incentives, and event rate on reaction time variability in children with ADHD. Neuropsychopharmacology : official publication of the American College of Neuropsychopharmacology36(5), 1060–1072. doi:10.1038/npp.2010.243

Hellwig-Brida, S., Daseking, M., Keller, F., Petermann, F., & Goldbeck, L. (2011). Effects of methylphenidate on intelligence and attention components in boys with attention-deficit/hyperactivity disorder. Journal of child and adolescent psychopharmacology21(3), 245-253.

Kempton, S., Vance, A., Maruff, P., Luk, E., Costin, J., & Pantelis, C. (1999). Executive function and attention deficit hyperactivity disorder: Stimulant medication and better executive function performance in children.Psychological Medicine, 29(3), 527-538. 

Semrud-Clikeman, M., Pliszka, S., & Liotti, M. (2008). Executive functioning in children with attention-deficit/hyperactivity disorder: Combined type with and without a stimulant medication history. Neuropsychology, 22(3), 329-340.

The literature review is mostly outdated. More recent literature is highly recommended to be included as this paper cites work that was conducted over 20 years ago. Especially work regarding neuroimaging and brain structure differences. There has also been more recent work on executive functions in ADHD, including executive functions not investigated in this paper (i.e., working memory).

The authors review some findings regarding a number of different executive functions (e.g., Barkley’s four and non-verbal working memory, inhibition, timing, and planning etc.) impaired in ADHD, but investigates mostly different executive functions (i.e., alertness, distractibility, divided attention, flexibility, and inhibition [go/no-go]). It would be more helpful if the authors describe previous literature of the executive functions that were investigated in the current study and explain what their study adds to the literature and/or why they chose the KiTAP battery to examine executive function deficits in medicated versus stimulant naive children with ADHD.

The authors should include specific hypothesis at the end of their introduction section.

Method:

The authors state that for the control group “the structured diagnostic interview must have confirmed the absence of ADHD.” Do the authors also exclude children that met criteria for other psychiatric disorders? Were children excluded if they met subthreshold criteria for any psychiatric disorder?

What were the qualifications (credentials) for the interviewers for the Mini-Kid? Who oversaw their work?

How did the authors exclude for intellectual disabilities? Did they test each child’s IQ?

The authors may want to state in their limitations section that their sample of ADHD children may not generalize to other groups of children with ADHD, as they excluded children with oppositional behavior, which is present in over half of the population of children with ADHD.

What were reasons for incomplete diagnostic interview?

The authors should describe each of their measures in detail so that the reader knows the exact parameters of the task.

The authors state the KiTAP has been used to measure EF in adults. Is it also used with children? This should be described in more detail.

Were all children in the medicated ADHD group taking their medication at the time of the KiTAP test? I’m assuming this was the case, but it was not stated directly.

Since the authors are performing a number of comparisons, they should control for family-wise error to decrease the risk of a Type I error. Were RTs from all trials used or just those from correct trials? Did the authors consider using ex-Gaussian measures of reaction time variability? Ex-Gaussian tau has been found to be a more reliable indicator of reaction time variability compared to standard deviation of reaction time in one study (see Epstein et al., 2011 mentioned above).

Results

17. The results section was a very confusing read to me. The figures and how they are used to describe the results were distracting. I am not sure what the figures add and there are too many of them. I recommend the authors (1) describe (in text) the main effects and significant covariates for each subtest (2) describe the post hoc tests for each comparison (as how they summarize the results in the discussion section) and (3) direct the reader to the tables for more detailed information.

18. The authors should include more details regarding medication type (i.e., how many children in the medication group received methylphenidate versus atomoxetine) and what was the average dosing was for the medication group.

19. What are the justifications for controlling for sport activity? The authors should also report this (as well as the data for the other control variables) in Table 1.

Discussion

20. The authors should update their literature review in the discussion section with more recent studies, including the one’s mentioned above.

21. The authors should mention that the medication naïve group may have had less severe symptoms from the beginning and not likely to require medication compared to the medication group.

Reviewer 2 Report

This paper provides an examination of the difference in the performance on EF tasks between children with ADHD symptoms on medication, children with symptoms not on medication, and children without symptoms and are not on medication.

The authors presented good support for their study and a sound approach to gathering the data needed. It is interesting to see that the results showed no significant difference between medicated children with symptoms of ADHD and typically developing children on 12 of the 15 KiTAP measures.

Additional comments/questions:

Regarding the analysis, it is not clear why previous treatment was not included as a control variable.

There is a more current version of the MINI Kid that is based on the DSM-5. Was this not available for your study?

Overall, I think the paper is excellent and would be a great addition to the literature.

Reviewer 3 Report

This manuscript addresses the important issue of executive functioning in ADHD in school age children. The authors investigate whether the presence of ADHD symptoms are linked to impaired of EF compared to typically developing children, and whether medication attenuates this association. The study was based on performance-based measures of EF.

Nevertheless, the ecological validity of the performance-based EF tests has been questioned.

There some crucial issues that have to be addressed in this study.

1. The ADHD diagnosis including the three scales: inattention, impulsivity hyperactivity

Have not been specified in this study.

Can the authors be sure that the symptoms they are studying can be attributed to ADHD? Are the authors sure about the ADHD diagnosis?

2. Quantification of the scores of this scales inattention, impulsivity hyperactivity) are needed in order to study groups differences in ADHD symptoms severity. Symptom severity have to be studied without medication, so a wash-up period before assessed symptoms is needed.

3. The homogeneity of the diagnosis between clinical subgroups: Are there differences in subtypes (inattentive, combined, hyperactive-impulsive) between the two medical conditions (medicated and unmedicated groups). These possible differences can interfere with the results.

4. Psychiatric symptoms: anxiety and depression symptoms accompanies ADHD and interfere with executive functions.

5. Are there between group differences in intellectual functioning?

6. The sample size is too small

7. Additionally, it is necessary to study age differences between groups because the age range is very large and the author reported age differences between groups.

8. Sex differences are also needed to be studied.

9. When studying group differences, it is also necessary in this case to assess the percentage of individuals in each group of subjects that have differences in executive functions (patients and controls).

10. In the discussion section, it is necessary to discuss together the data corresponding to EF functions and the clinical diagnosis of ADHD.

Round 2

Reviewer 1 Report

I appreciate the authors responding to each of my comments, I do think that they have sufficiently addressed all of them. I have two additional suggestions. The first one concerns the readability of the paper, particularly in the introduction and discussion sections  First while the authors did provide a more updated, complete, and balanced literature review; it was, at times, hard to follow.  The authors describe Barkley's model of executive function, and spend some time discussing executive functions that were not investigated in the current study (e.g., reconstitution, working memory, internalization of speech). I think that they could mention Barkley's theory and give brief evidence that those EFs are impaired in ADHD (two or three sentences), but then discuss evidence for the EFs they are investigating in their study. For example, Lines 45-70 and 77-103 could be combined into one or two paragraphs and summarized concisely. There are also a number of grammatical errors throughout the manuscript. Second, the authors should not be interpreting their  non-significant findings in the discussion section (e.g., Alertness median of reaction time, Go-nogo median reaction time).
